# Associations between Somatic Cell Count and Milk Fatty Acid and Amino Acid Profile in Alpine and Saanen Goat Breeds

**DOI:** 10.3390/ani13060965

**Published:** 2023-03-07

**Authors:** Evaldas Šlyžius, Lina Anskienė, Giedrius Palubinskas, Vida Juozaitienė, Birutė Šlyžienė, Renalda Juodžentytė, Lina Laučienė

**Affiliations:** 1Department of Animal Breeding, Faculty of Animal Science, Lithuanian University of Health Sciences, Tilžes 18, LT-47181 Kaunas, Lithuania; 2Department of Biology, Faculty of Natural Sciences, Vytautas Magnus University, K. Donelaicio 58, LT-44248 Kaunas, Lithuania; 3Consulting and Study Centre for Postgraduate Studies in Veterinary Sciences, Lithuanian University of Health Sciences, Tilžes 18, LT-47181 Kaunas, Lithuania; 4Department of Food Safety and Quality, Faculty of Veterinary Medicine, Lithuanian University of Health Sciences, Tilžes 18, LT-47181 Kaunas, Lithuania

**Keywords:** somatic cell count, fatty acid, amino acid, goat, milk

## Abstract

**Simple Summary:**

Although somatic cell count is not a sensitive biomarker to identify mastitis in goats, the current study revealed a relationship between this indicator and goat milk composition. While data on goat milk fatty acids and especially on amino acid variations according to somatic cell count influence are very minimal, the main goal of this study was to evaluate the relation of different somatic cell count levels in goat milk with goat milk yield, milk composition, fatty acid, and amino acid profiles of Alpine and Saanen goat breeds. The research revealed some statistically significant relationships between somatic cell count and amino acids and fatty acids, suggesting that improving milk quality by reducing somatic cell count may benefit farmers by improving goat milk’s fatty acid and amino acid composition and may serve as a biomarker for dairy goats. However, future studies with more goats are needed to confirm these results.

**Abstract:**

The main goal of this study was to evaluate the relation of different SCC levels in goat milk with goat milk yield, milk composition, FA, and AA profiles. Whereas the investigated herd was composed of Alpine and Saanen goats, the influence of breed on milk parameters and milk yield was also assessed. The research was carried out in 2022 at a Lithuanian dairy goat farm with 135 goats (Saanen = 66 and Alpine = 69) without evidence of clinical mastitis. The current research revealed a relationship between SCC with goat milk yield and composition. Goats with a high SCC had significantly lower milk yield (*p* < 0.001), lower content of lactose (*p* < 0.01), fat (*p* < 0.001) and higher protein content (*p* < 0.05) in their milk. The increase in most AA was significantly associated with increased SCC. The higher quantity of Asp, Glu, Ala, Met, His, Lys, Arg, EAA, NEAA, and TAA (compared with the low SCC group) (*p* < 0.05–0.01), Leu, Tyr, and BCAA (compared with the low and medium SCC group) were found in the milk of the high SCC group (*p* < 0.05–0.01). The distribution of the main FA groups was also related to SCC and showed a significant decrease in SCFA (*p* < 0.01–0.001) and an increase in LCFA, PUFA, and BCFA in the high SCC group (*p* < 0.05). All individual AA and their groups (EAA, NEAA, TAA, BCAA) were significantly lower in the milk of the Saanen goat breed (*p* < 0.001). The most individual FA ranged between goat breeds, while the total amount of SFA, UFA, and MUFA wasn’t affected by breed (*p* > 0.05). The research revealed a statistically significant relationship between SCC, AA, and FA, suggesting that these traits may be used as a biomarker in the goat selection process.

## 1. Introduction

Milk is one of the most valuable and widely used animal-origin products. Although 80% of dairy products are made from cow’s milk [1], goat’s milk production is highly valued due to its nutritive values and positive health benefits attached to these products [2].

Goat milk differs in some physicochemical properties from cow’s milk. The β-casein/αs1-casein ratio (70/30%) of goat milk proteins is similar to human milk, while αs1-casein is the primary casein in cow milk. Goat milk has significant effects on hypo allergenicity and higher digestibility than cow milk because β-casein is more sensitive to the action of pepsin than αs1-casein [2]. Although the amino acid (AA) profile of cow and goat milk proteins coincides by around 90% [3], the protein fraction of goat milk has a higher concentration of six (threonine, isoleucine, lysine, cystine, tyrosine, and valine) out of the ten essential amino acids present compared with bovine milk [4]. Goat milk usually contains 18 of the 20 amino acids, except for two—Asparagine (Asn) and Glutamine [5]. However, the reason for this may be related to the possibility of the differentiating of two closely related amino acids; therefore, some authors calculate Asparagine with Aspartic acid and Glutamine with Glutamic acid and label them as Asx and Glx, respectively [6].

Lipids of goat milk provide better digestibility with small fat globule sizes (3.19 to 3.50 µm) and are characterized by a high concentration of short and medium-chain fatty acids (SCFA and MCFA) synthesized de novo in the mammary gland. SCFA and MCFA contribute to goat milk flavor and, together with unsaturated fatty acids (UFA), have beneficial effects such as antimicrobial or antithrombotic on human health [2,7,8,9]. 

Production and quality of dairy goat milk can be affected by various factors such as nutrition, breed, lactation stage, parity, environment, season, and certainly udder health status [10]. An increase in somatic cell counts (SCC) may be associated with a mammary gland health condition and relate to milk yields and composition [11]. In addition to protective functions in milk, SCC also provides numerous enzymes and may affect dairy processes and the quality of products [12]. Although high SCC is strongly associated with mastitis in cows, that is not always the case with goats. Generally, it is agreed that goat milk has a higher SCC than cow milk, and the interpretation of SCC in goats is more complicated than in cows [13].

Despite the absence of legal SCC regulation for goat milk in the EU, research on this subject is performed. But most of the studies on goat milk focus on essential milk components such as fats, proteins, lactose, and milk yield [12,13,14], while data on goat milk FA and especially on AA variation according to SCC influence is very minimal. Sramek et al. reported that udder health significantly affected the concentration of SCFA and MCFA (from C4 to C14) in goat milk. These FA decreased with an increase in SCC in goat milk. In contrast, the long-chain fatty acids (LCFA), which originate from blood throughout triglyceride mobilization from the small intestine (90%) or the adipose tissue (10%), increased with the increase in SCC in goat milk [15]. Meanwhile, data on goat milk AA composition under SCC impact is extremely limited. 

Therefore, the main goal of this study was to evaluate the relation of different SCC levels in goat milk and goat milk yield, milk composition, FA, and AA profiles of Alpine and Saanen goat breeds.

## 2. Material and Methods

### 2.1. Animals, Sampling and Feeding

This study was performed in 2022 at a Lithuanian dairy goat farm, from June to August. 135 goats (Saanen, N = 66, and Alpine, N = 69) without evidence of clinical mastitis were assigned to 3 groups according to SCC level of <750 × 103 cell/mL (low level group, N = 60), 750–1000 × 103 cell/mL (medium group, N = 30), and >1000 × 103 cell/mL (high level group, N = 45) [16].

All investigated goats, weighing from 40 to 47 kg, were on average 3.00 ± 0.13 parity (2.97 ± 0.12 Saanen, 3.02 ± 0.13 Alpine), on a middle stage of lactation (90 ± 7.08 days—Saanen, 92 ± 7.15—Alpine) at the start of the experiment, and were raised in the same feeding and housing conditions. The animals received a total mixed ration from pasture grass and hay (ad libitum) and concentrates (600 g) (Table 1). Goats had access to water ad libitum.

Goats were milked twice a day, at 7:00 a.m. and 6:00 p.m. The milking parlour had a low-line design with self-locking gates and two platforms consisting of eight milking units and milking posts per platform. Individual milk samples were collected three times during the experiment, from June to August (on the 20th day of each month). 

### 2.2. Analysis of Milk Composition and SCC

Analysis of goat milk fat, protein, lactose, and urea was made using spectrophotometers LactoScope 550 and LactoScope FTIR (Delta Instruments, Drachten, The Netherlands) by the JSC “Pieno Tyrimai” laboratory, which is accredited by the National Accreditation Bureau to perform physical-chemical and microbiological tests of raw milk. 

The somatic cell count in milk was determined using the measuring device Somascope CA-3A4 (Delta Instruments, Drachten, The Netherlands), which operates on the principle of the flow cytometry method. Goat milk yield was analyzed during control milking and evaluated according to the data from the Agricultural Information and Rural Business Centre. 

### 2.3. Analysis of the Composition of Fatty Acids and Amino Acids

The composition of fatty acids (FA) and amino acids (AA) was determined at the Chemical Laboratory of the Livestock Farming Institute of the Lithuanian University of Health Sciences. 

4000 rpm centrifugation separated the cream from the goat milk. Fats were collected using a 3:1 blend of chloroform and methanol and methylated with a 2% sodium methylate [17]. FA methyl esters were injected into the CG-2010 SHIMADZU gas chromatographer integrated with a hydrogen flame detector. The FA were recognized based on the output times of a known standard (Supelco 37 FAME mix, Linoleic acid methyl ester isomer mix, Supelco Trans FAME mix K110) and measured by the CG Solution data processing program. Single FA were expressed as a percentage of the total FA and were categorized into short-chain fatty acids (SCFA, C4–C8), medium-chain fatty acids (MCFA, C10–C15), and long-chain fatty acids (LCFA, C16, and more) by the number of carbon atoms [18]. Furthermore, FA were defined into saturated fatty acids (SFA), unsaturated fatty acids (UFA), monounsaturated fatty acids (MUFA), polyunsaturated fatty acids (PUFA), and branched-chain fatty acids (BCFA) based on the number of single and double bonds or methyl branches on the FA chain.

Before the AA evaluation, milk samples were degreased by centrifugation at 4000 rpm. The degreased samples were hydrolyzed in compliance with Commission Regulation (EC) No 152/2009 [19], while derivatization and quantification were completed following the Waters AccQ Tag Chemistry Package Instruction Manual. AA were expressed as g/kg of degreased milk. Histidine (His), isoleucine (Ile), leucine (Leu), lysine (Lys), methionine (Met), phenylalanine (Phe), threonine (Thr), and valine (Val) were grouped into essential amino acids (EAA); alanine (Ala), arginine (Arg), aspartic acid (Asp), glutamic acid (Glu), glycine (Gly), proline (Pro), serine (Ser), and tyrosine (Tyr) were grouped into nonessential amino acids (NEAA). Additionally, the total content of AA (TAA), branched-chain amino acids (Leu + Ile + Val, BCAA), and the ratio of EAA/TAA were counted.

### 2.4. Statistical Analysis

Statistical data analysis was performed using SPSS 25.0 analytical software. The data were presented using descriptive statistics. Normal distributions of variables were assessed using the Kolmogorov–Smirnov test. The one-way analysis of variance (ANOVA) was used to determine statistically significant differences between the means. Multiple comparisons of groups means were calculated using the Post—Hoc Tukey test. The differences were considered significant at *p* < 0.05.

## 3. Results and Discussion

### 3.1. Milk Yield and Composition in Alpine and Saanen Goats Depending on the Level of SCC in Milk

The goat milk yield and composition according to goat breed and SCC group are presented in Table 2. The present study showed that Saanen goats yielded (*p* < 0.001) more milk (2.45 ± 0.04 vs. 1.71 ± 0.02 kg/day), but their milk was significantly lower in fat (3.13 ± 0.10 vs. 4.39 ± 0.13%) and protein (3.02 ± 0.04 vs. 3.53 ± 0.05%) content than that of Alpine goats [20]. Lower fat and protein content with higher milk yield in Saanen goats may be attributed to goat genetics and milk dilution effects [20]. A few previous studies revealed similar milking results: Saanen goats had significantly higher daily milk yields (2.15–2.63 kg/day) compared with Alpine goats (1.76–2.08 kg/day) [21,22], while milk composition varied differently between investigated breeds. Mioč et al. [21] found that Alpine milk was richer in fat content than Saanen milk (3.47 vs. 3.25 %) but the protein and lactose content did not differ significantly between breeds. Meanwhile, Shuvarikov et al. [22] found significantly higher protein content in the milk of Alpine goats than in Saanen goats (3.72 vs. 3.55%), while fat and lactose content remained similar in both breeds. Costa et al. [4] estimated that protein and lactose were higher 12.50 and 3.40%, respectively, in the milk of Alpine goats, but fat content did not differ significantly between breeds. 

The current study showed that the amount of urea was significantly higher (*p* < 0.05) in Alpine milk than in Saanen milk, while lactose did not differ significantly under the breed influence. Milk urea shows whether the diet of ruminants is well balanced by energy and protein; however, in goat’s milk, unlike cow’s, it is not controlled Bendelja-Ljoljić [23], Brun-Bellut et al. [24] reported an optimum milk urea range of 28–32 mg/dL for dairy goats. SCC was higher in the milk of Alpine goats compared with Saanen breed goats.

In the current study, the average SCC in the milk of the high SCC group was 9.1 times higher (2201 ± 183 × 10^3^ cell/mL) than in low SCC (242.5 ± 14.8 × 10^3^ cell/mL) group milk (*p* < 0.001), (Table 2). Although in the present study, none of the investigated goats had visual symptoms of mastitis, some of those assigned to a high SCC group could suffer from subclinical mastitis. Previous studies, where the milk of uninfected and infected goat glands/halves was analyzed, estimated similar results for the SCC: 350 vs. 1600 × 10^3^ cell/mL [25] or 417 vs. 1750 × 10^3^ cell/mL) [12]. On the other hand, SCC in goat milk can range from 270 × 103 to 2000 × 103 cells/mL without mastitis, according to Paape et al. [26]. Previous studies have shown that goat milk has an average of 150 × 103/mL of cytoplasmic particles, which could be mistakenly counted as somatic cells. Therefore, only cell counting procedures specific to DNA should be used to obtain accurate SCC for goats [26,27]. Milk yield, fat, and lactose content decreased (9.68%, *p* < 0.001; 21.68%, *p* < 0.001; 6.18%, *p* < 0.01), while protein content increased (12.34%, *p* < 0.05) significantly when SCC exceed 1000 × 10^3^ cell/mL in goat milk compared with the milk of low SCC group. Javier [28] estimated that the percentage losses of goat milk yield for SCC levels of 1000 ×10^3^ cells/mL, 2000 × 10^3^ cells/mL, 3000 × 10^3^ cells/mL and >7000 × 10^3^ cells/mL were 11.4, 19.5, 24.2, and 35.7%, respectively. Cedden et al. [29]. also found a negative relationship between milk yield and higher SCC in goats without clinical mastitis signs. Akers and Nickerson [30] state that a decrease in milk yield can be related to damage to the udder parenchyma and blockage of milk ducts with bacteria, neutrophils, and fibrin. Besides, Koop et al. [31] found that the decrease in milk yield during subclinical mastitis was different for various pathogen groups. Barrón Bravo et al. [5] found that goats with high SCC had the lowest milk yield and fat content and the highest protein content in their milk, which coincided with the present study’s data. Other studies showed that fat content did not differ, protein content increased significantly or tended to increase, and lactose decreased when SCC exceeded 1000 × 10^3^ cell/mL in the milk of healthy goats [11] or 1750 × 10^3^ cell/mL in the milk of infected goats [12]. Meanwhile, Sramek et al. [15] found a non-significant increase in proteins, a significant increase in milk fat, and a significant decrease in lactose content in the high SCC group (>1000 × 10^3^ cell/mL) milk of Alpine goats. The reduction in lactose concentration can be a result of increased plasmin activity; the higher content of total protein can be related to the increase in albumin and whey protein concentrations in milk during subclinical or clinical mastitis [32].

### 3.2. Amino Acids Composition 

The quantity of all individual AA and their groups (EAA, NEAA, TAA, and BCAA) was significantly lower in Saanen compared with Alpine goat milk, *p* < 0.001 (Figure 1). Only the ratio of EAA/TAA was higher (0.45 ± 0.01 vs. 0.46 ± 0.01) of Saanen goat milk (*p* < 0.001).

Despite the mentioned inter-breed differences, Glu, following Pro, Leu, Lys, and Asp, had the highest concentration (~ 20, ~10, ~9, ~8, and ~7%), in both breed’s milk (Figure 2), and these AA values were in agreement with Landi et al. [6]. Meanwhile, the contents of Gly, Arg, Ala, His, and Tyr were found in the lowest range, which was 1.90–4.20% for Alpine milk and 1.78–3.79% for Saanen milk.

Other studies showed that high levels of Pro, Leu, and Glu + Gln were determined in goat milk from Saanen and Alpine goat breeds, while His and Tyr were found to be at a lower level than the other amino acids determined in goat milk samples. It was found that the contents of Pro, Leu, and Glu + Gln were in the range of 25.49–50.14, 24.53–35.79, and 21.34–35.28 µmol/mL, respectively, while the contents of His and Tyr were found to be in the range of 3.70–5.36 and 3.31–5.67 µmol/mL, respectively [32].

Most of the AA were significantly increased (Table 3) when SCC reached over 1000 × 10^3^ cells/mL in goat’s milk. The higher quantity of Asp, Glu, Ala, Met, His, Lys, Arg, EAA, NEAA, TAA (compared with the low the SCC group) Leu, Tyr, and BCAA (compared with the low and medium SCC group) were found in the milk of the high SCC group (*p* < 0.05–0.01), while the milk of the medium SCC group had a higher content of Thr, Ile, and Phe (compared with the low SCC group), (*p* < 0.05–0.01) and the ratio of EAA/TAA (compared with the low and high SCC group), (*p* < 0.05). The higher content of individual and TAA can be related to the increased albumin and whey protein concentrations in milk during subclinical or clinical mastitis [12]. Although albumin and whey proteins have not been analyzed in the present study, the increase in total protein content in milk with high SCC can be associated with the rise in those milk components.

Data on the SCC impact on the AA composition of goat milk are lacking, but a study with cows revealed similar results. Andrei et al. showed that total AA content increased significantly in cow mastitis milk compared with normal (12,073.06 ± 5564.14 µg/mL vs. 619.82 ± 76.02 µg/mL). Also, an increase in Lys, Tyr, Leu, Ile, Val, Gly, and Ala, while a decrease in Glu, Asp, Met, Pro, Ser, and Thr was observed in the milk of cows with mastitis symptoms [33].

### 3.3. Fatty Acids Composition

The FA composition of goat milk of the different breeds is presented in Table 4. The highest concentrations of FA found in milk were C10:0, C14:0, C16:0, C18:0, and C18:1n9c. These FA accounted for ~78% of the total for both goat breeds, which is consistent with findings reported by Kuchtik et al. [10] and Mohsin et al. [34]. Nevertheless, most individual FA varied between goat breeds.

Significantly less of C8:0, C10:0, C11:0, C12:0, C14:0, C16:1n7, C17:1n9, C18:2n6t, C18:2n6c,t, C18:2n6t,c, C18:3n3, C20:5n3, C22:4n6, and C22:5n3 whereas more of C4:0, iC14:0, aC15:0, iC17:0, C18:0, C20:0, and C20:1n9 were found in the Saanen compared with the Alpine milk. Vulić et al. [8] also revealed variation of FA in the milk of Alpine and Saanen breeds, however, goats were on different breeding regimes. Earlier studies confirmed FA variation between different goat breeds such as Bulgarian White Dairy Goat and Toggenburg (Pamukova et al. [35]) or Nguni and Boer (Idamokoro et al. [36]) when goats were kept in similar conditions. 

Although a significant variation in single FA was found, the total amount of SFA, UFA, and MUFA did not differ between breeds (Figure 3), and their estimated percentages were in line with the literature (Mollica et al. [37]). This study showed that the amounts of LCFA (3.21%) and BCFA (26.14%) was higher in the milk of Saanen goats and lower in MCFA (10.27%) and PUFA (7.52%) compared with Alpine goat milk. Vulić et al. [8] did not reveal variation in SFA and PUFA, although goats being kept under different breeding models, but found a significant difference in the MUFA content between Alpine (22.80 g/100 g) and Saanen (24.00 g/100 g) goats kept in Bosnia and Herzegovina [31]. According to most studies, the proportion of MUFA in the total FA goat’s milk content can vary from 15 to 30%, and the intrinsic content of oleic acid (C18:1n9c) is the main determining factor (Kuchtik et al. [10]), which coincides with the data of the current study.

The content of BCFA estimated in the milk of both goat breeds was from 2.16 to 2.70 times lower than that estimated by Lopez et al. [38]. Although BCFA is a lesser constituent of milk, it revealed a positive effect on gastrointestinal microbiota and can act as a potential anti-cancer agent [38]. Our analysis revealed that in the milk of the Alpine goat there was significantly more omega-3 FA than in the milk of Saanen (1.01 ± 0.18 g/100 g vs. 0.72 ± 0.13 g/100 g) in the present study; consequently, the ratio of omega-6/omega-3 was also more favorable (2.89 vs. 4.05) in Alpine milk. Due to a higher dietary intake of omega-6 in recent years, the ratio of omega-6/omega-3 FA increased to 15–20:1 in the human diet, while the ideal balance must be around 2:1 [39]. Values of omega-6 and omega-3 FA and their proportion in our study are similar to those recorded by Ceballos et al. [40] in goat milk, while milk from cows’ had a markedly higher ratio of omega-6/omega-3, precisely 10.49. 

The distribution of the main FA groups according to SCC groups is presented in Figure 4. A significant decrease in SCFA and an increase in LCFA, PUFA, and BCFA were found when SCC exceeded 1000 × 10^3^ cells/mL in goat milk, but our data analysis revealed a weak positive (r = 0.251, *p* < 0.01) correlation only between SCC and PUFA. The data obtained in this study is similar to Sramek et al. [15]. This author estimated that SCFA and MCFA (from C4 to C14) were lower, and in contrast, the LCFA concentrations were higher in goat milk with a high SCC (<750 × 10^3^ cell/mL vs. >1000 × 10^3^ cell/mL).

SCFA, MCFA, and partially C16:0 are de novo synthesized in epithelial cells of the mammary gland from rumen acetate and butyrate, while diet lipids are the precursors for LCFA and they are passed over from the blood to the goat mammary gland [41]. FA’s de novo synthesis pathway mainly depends on Acetyl-CoA carboxylase (ACACA) and fatty acid synthase (FASN) expression. Pro-inflammatory cytokines released in infected mammary glands reduce the expression of FASN and ACACA [42]. Yakan et al. [43] estimated a significant negative (r = − 0.407, *p*  <  0.001) correlation between SCC and FASN as well. In the current study, some goats in a high SCC group with an average of 2201 ± 183×10^3^ cells/mL could suffer from subclinical mastitis; thus, SCFA synthesis may have been disturbed in their udder parenchyma.

Corresponding changes under the SCC group were also seen in the content of individual FAs (Table 5). C4:0, C6:0, C8:0, iC15:0, C16:1n7, iC17:0, C20:0, and C20:1n9 decreased, whereas C13:0, iC14:0, aC15:0, C16:1n9t, C17:1n9, C18:1n7, C18:2n6tc, and C20:4n6 increased when SCC reached over 1000 × 10^3^ cells/mL, compared with the low-SCC or medium-SCC groups. 

However, the individual FA, such as C10:0, C14:0, C16:0, C18:0, and C18:1n9, which comprise around 78% of total FA, were not affected by SCC in goat milk. In contrast, Sramek et al. [15] found that C16:0 concentration was lower, while C18:1n9 and simultaneously MUFA concentrations were higher in the milk of the high SCC group. The author assumes that the increased concentration of all 18-carbon and longer chain fatty acids in milk with high SCC may be related to the decreased substrate competition with SCFA and MCFA during milk fat synthesis.

Although SCC is not a sensitive biomarker to identify mastitis in goats, the current study revealed a relationship between this indicator and goat milk yield and composition. Goats with a high SCC had significantly lower milk yield, lactose, and fat content in their milk. In contrast, protein content, including most of AA, increased with increased SCC in goat milk. The higher quantity of Asp, Glu, Ala, Met, His, Lys, Arg, EAA, NEAA, TAA (compared with the low SCC group), Leu, Tyr, and BCAA (compared with the low and medium SCC groups) were found in the milk of the high SCC group. Analysis of FA composition showed a decrease in SCFA and an increase in BCFA, LCFA, and PUFA in milk with a high SCC group.

All individual AA and their groups (EAA, NEAA, TAA, BCAA) were significantly lower in the milk of the Saanen goat breed. Most individual FA ranged between goat breeds, while the total amount of SFA, UFA, and MUFA wasn’t significantly affected by goat breed.

## 4. Conclusions

The research revealed a statistically significant association of SCC with AA and FA, suggesting that improving milk quality by SCC may be beneficial to farmers in improving the fatty acid and amino acid composition of goat milk and may possibly serve as a biomarker in the selection of dairy goats; however, future studies with a larger number of goats are needed to confirm these results.

## Figures and Tables

**Figure 1 animals-13-00965-f001:**
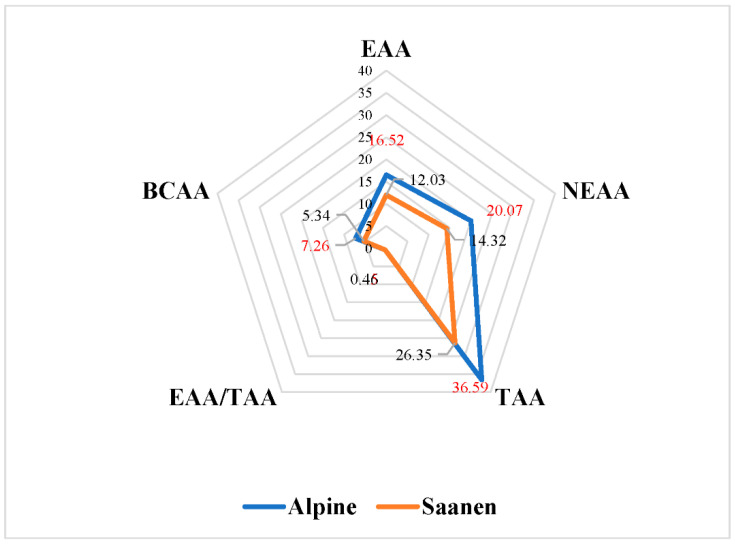
The average of amino acids (g/kg of degreased milk) groups in milk of Alpine and Saanen goat breeds. Essential amino acids—EAA; nonessential amino acids—NEAA; total amino acids—TAA; branched-chain amino acids (Leu + Ile + Val, BCAA); ratio of EAA/TAA—EAA/TAA.

**Figure 2 animals-13-00965-f002:**
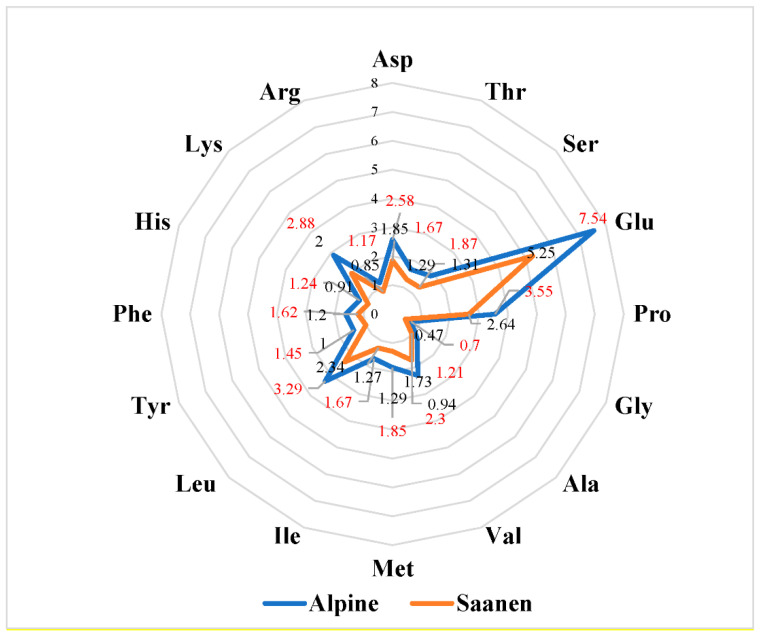
The average of amino acids (g/kg of degreased milk).

**Figure 3 animals-13-00965-f003:**
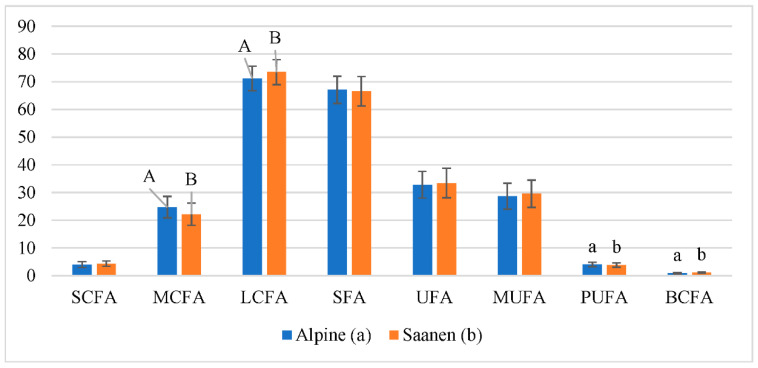
The distribution of the main fatty acid groups of Alpine and Saanen goat breeds. Means with different superscripts within the same line represented differ significantly at *p* < 0.01 (A, B) or *p* < 0.05 (a, b) level. Short-chain fatty acids—SCFA; medium-chain fatty acids—MCFA; long-chain fatty acids—LCFA; saturated fatty acids—SFA); unsaturated fatty acids—UFA; monounsaturated fatty acids—MUFA; polyunsaturated fatty acids—PUFA; branched-chain fatty acids—BCFA.

**Figure 4 animals-13-00965-f004:**
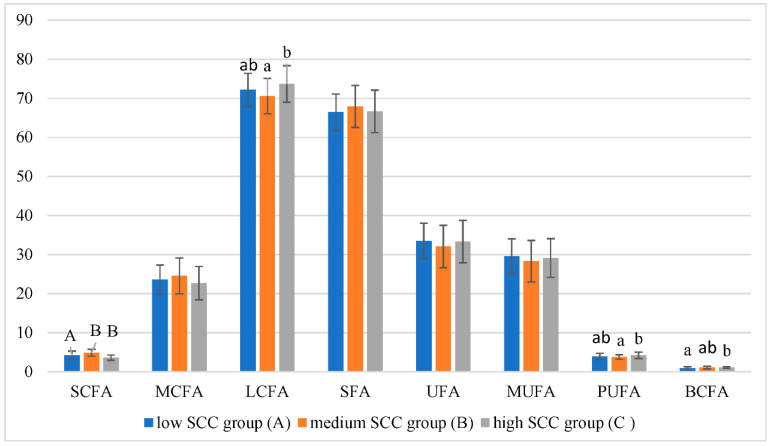
The distribution of the main fatty acid groups according to SCC groups. Means with different superscripts within the same line represented differ significantly at *p* < 0.01 (A, B) or *p* < 0.05 (a, b) level. Short-chain fatty acids—SCFA; medium-chain fatty acids—MCFA; long-chain fatty acids—LCFA; saturated fatty acids—SFA); unsaturated fatty acids—UFA; monounsaturated fatty acids—MUFA; polyunsaturated fatty acids—PUFA; branched-chain fatty acids—BCFA.

**Table 1 animals-13-00965-t001:** The components of the total mix ration (TMR). Daily feeding for each animal.

Indicators	Pasture Grass	Hay	Concentrate
Dry matter, g/kg	247	830	855
Ash, g/kg	83	61	20
Crude protein, g/kg	123	111	103
Crude fibre, g/kg	231	242	51
Crude fat, g/kg	33	22	21
Total sugar, g/kg	117	130	52
Acid detergent fibre (ADF), g/kg	293	350	-
Acid detergent lignin, g/kg	45	415	-
Neutral detergent fibre (NDF), g/kg	447	486	182
Net energy for lactation (NEL), MJ	6.2	8.6	8.3
Digestibility of organic matter (%, OM)	71	60	85.7

**Table 2 animals-13-00965-t002:** Milk yield, composition and SCC of Alpine and Saanen goat breeds.

Indicators	Breed		SCC Group (10 × 1000 cell/mL)		
			Low	Medium	High
	Saanen	Alpine	<750	750–1000	>1000
	N = 66	N = 69	N = 60	N = 30	N = 45
Milk yield, kg/per day	2.45 ± 0.04 ^A^	1.71 ± 0.02 ^B^	2.17 ± 0.06 ^A^	2.16 ± 0.12 ^AB^	1.96 ± 0.04 ^B^
Fat, %	3.13 ± 0.10 ^A^	4.39 ± 0.13 ^B^	4.29 ± 0.17 ^A^	3.91 ± 0.37 ^AB^	3.36 ± 0.10 ^B^
Protein, %	3.02 ± 0.04 ^a^	3.53 ± 0.05 ^b^	3.08 ± 0.04 ^a^	3.45 ± 0.09 ^b^	3.46 ± 0.07 ^b^
Lactose, %	4.14 ± 0.03	4.09 ± 0.03	4.21 ± 0.02 ^A^	4.05 ± 0.04 ^B^	3.95 ± 0.03 ^B^
Urea, mg/dL	26.67 ± 1.03 ^a^	29.53 ± 1.41 ^b^	26.82 ± 1.09	29.72 ± 3.31	27.26 ± 1.47
SCC, ×10^3^ cells/mL	718.7 ± 64.8 ^a^	1085 ± 162 ^b^	242.5 ± 14.8 ^A^	845.6 ± 23.7 ^B^	2201 ± 183 ^C^

Means with different superscripts within the same line represented differ significantly at *p* < 0.01 (A, B, C) or *p* < 0.05 (a, b) level.

**Table 3 animals-13-00965-t003:** The average of amino acids and their groups according to SCC groups.

AA, g/kg of Degreased Milk	SCC Group (10 × 1000 cell/mL)		
	Low	Medium	High
	<750	750–1000	>1000
	N = 60	N = 30	N = 45
Aspartic acid (Asp)	2.01 ± 0.43 ^A^	2.29 ± 0.56 ^AB^	2.35 ± 0.64 ^B^
Threonine (Thr)	1.37 ± 0.26 ^a^	1.62 ± 0.34 ^b^	1.51 ± 0.43 ^ab^
Serine (Ser)	1.48 ± 0.33	1.64 ± 0.39	1.66 ± 0.55
Glutamic acid (Glu)	5.80 ± 1.31 ^A^	6.59 ± 1.56 ^AB^	6.80 ± 1.83 ^B^
Proline (Pro)	2.91 ± 0.53	3.14 ± 0.63	3.32 ± 0.87
Glycine (Gly)	0.54 ± 0.13	0.65 ± 0.21	0.60 ± 0.26
Alanine (Ala)	1.03 ± 0.15 ^a^	1.04 ± 0.14 ^ab^	1.14 ± 0.27 ^b^
Valine (Val)	1.89 ± 0.34	2.09 ± 0.46	2.08 ± 0.56
Methionine (Met)	1.41 ± 0.33 ^A^	1.60 ± 0.41 ^AB^	1.69 ± 0.58 ^B^
Isoleucine (Ile)	1.35 ± 0.27 ^A^	1.61 ± 0.35 ^B^	1.49 ± 0.43 ^AB^
Leucine (Leu)	2.54 ± 0.56 ^a^	2.95 ± 0.68 ^b^	2.98 ± 0.81 ^b^
Tyrosine (Tyr)	1.11 ± 0.25 ^a^	1.31 ± 0.38 ^b^	1.28 ± 0.37 ^b^
Phenylalanine (Phe)	1.31 ± 0.26 ^a^	1.52 ± 0.37 ^b^	1.44 ± 0.41 ^ab^
Histidine (His)	1.01 ± 0.20 ^a^	1.05 ± 0.20 ^ab^	1.14 ± 0.29 ^b^
Lysine (Lys)	2.22 ± 0.51 ^A^	2.48 ± 0.55 ^AB^	2.61 ± 0.76 ^B^
Arginine (Arg)	0.95 ± 0.20 ^A^	0.98 ± 0.19 ^AB^	1.09 ± 0.24 ^B^
EAA	13.10 ± 2.69 ^a^	14.91 ± 3.23 ^ab^	14.95 ± 4.13 ^b^
NEAA	15.82 ± 3.30 ^a^	17.63 ± 3.86 ^ab^	18.15 ± 4.89 ^b^
TAA	28.92 ± 5.98 ^a^	32.54 ± 7.07 ^ab^	33.10 ± 9.01 ^b^
BCAA	5.78 ± 1.16 ^a^	6.65 ± 1.46 ^b^	6.56 ± 1.76 ^b^
EAA/TAA	0.45 ± 0.01 ^a^	0.46 ± 0.01 ^b^	0.45 ± 0.01 ^a^

Means with different superscripts within the same line represented differ significantly at *p* < 0.01 (A, B) or *p* < 0.05 (a, b) level. AA—amino acids; essential AA (EAA), nonessential AA (NEAA), total AA (TAA), branched-chaiAA (Leu + Ile + Val, BCAA) AA; EAA/TAA—the ratio of EAA and TAA.

**Table 4 animals-13-00965-t004:** The average of fatty acids and their groups in milk of Alpine and Saanen goat breeds.

FA, g/100 g of Total FA	Breed	
	Alpine	Sannen
	N = 69	N = 66
Butyric acid (C4:0)	0.94 ± 0.78 ^A^	1.47 ± 0.78 ^B^
Caproic acid (C6:0)	1.02 ± 0.20	1.03 ± 0.19
Caprylic acid (C8:0)	2.02 ± 0.43 ^A^	1.81 ± 0.40 ^B^
Capric acid (C10:0)	8.46 ± 1.77 ^A^	7.28 ± 1.75 ^B^
Undecanoic acid (C11:0)	0.23 ± 0.10 ^A^	0.19 ± 0.04 ^B^
Lauric acid (C12:0)	4.20 ± 0.97 ^A^	3.38 ± 1.04 ^B^
Tridecanoic acid (C13:0)	0.15 ± 0.12	0.16 ± 0.10
Myristic acid (C14:0)	9.86 ± 1.47 ^A^	9.12 ± 1.42 ^B^
iso C14:0	0.45 ± 0.21 ^A^	0.57 ± 0.15 ^B^
Myristoleic acid (C14:1n7)	0.20 ± 0.11	0.17 ± 0.20
Pentadecylic acid (C15:0)	0.86 ± 0.16	0.93 ± 0.34
iso C15:0	0.09 ± 0.08	0.06 ± 0.10
anteiso C15:0	0.22 ± 0.08 ^A^	0.32 ± 0.15 ^B^
Palmitic acid (C16:0)	26.75 ± 2.80	26.81 ± 3.67
Trans-palmitoleic acid (C16:1n9 t)	0.58 ± 0.18	0.55 ± 0.14
Palmitoleic cis 9 acid (C16:1n9)	0.90 ± 0.15	0.87 ± 0.16
Palmitoleic acid (C16:1n7)	0.68 ± 0.21 ^A^	0.57 ± 0.11 ^B^
Margaric acid (C17:0)	0.93 ± 0.29	0.85 ± 0.32
iso C17:0	0.11 ± 0.12 ^a^	0.15 ± 0.09 ^b^
Margarineoleic acid (C17:1n9)	0.44 ± 0.21 ^A^	0.34 ± 0.11 ^B^
Stearic acid (C18:0)	9.76 ± 2.73 ^A^	11.40 ± 1.72 ^B^
Elaidic acid (C18:1n9t)	1.70 ± 0.72	1.72 ± 0.69
Oleic acid (C18:1n9)	23.04 ± 4.25	24.33 ± 4.59
Vaccenic acid (C18:1n7)	0.98 ± 0.27	0.94 ± 0.23
Linolelaidic acid (C18:2n6t)	0.38 ± 0.17 ^a^	0.33 ± 0.13 ^b^
C18:2n6c,t	0.24 ± 0.10 ^a^	0.21 ± 0.08 ^b^
C18:2n6t,c	0.26 ± 0.17 ^a^	0.21 ± 0.07 ^b^
Linoleic acid (C18:2n6c)	2.04 ± 0.55	2.17 ± 0.50
Alpha linolenic acid (C18:3n3)	0.76 ± 0.30 ^A^	0.55 ± 0.18 ^B^
Eisocanoic acid (C20:0)	0.22 ± 0.08 ^A^	0.30 ± 0.13 ^B^
Gondoic acid (C20:1n9)	0.06 ± 0.07 ^A^	0.11 ± 0.04 ^B^
Arachidonic acid (C20:4n6)	0.16 ± 0.06	0.17 ± 0.06
Eicosapentaenoic acid (C20:5n3)	0.06 ± 0.06 ^A^	0.02 ± 0.04 ^B^
Heneicosylic acid (C21:0)	0.73 ± 0.72	0.65 ± 0.31
Behenic acid (C22:0)	0.09 ± 0.11	0.09 ± 0.06
Dosocatetraenoic acid (C22:4n6)	0.03 ± 0.08 ^a^	0.00 ± 0.00 ^b^
Docosapentaenoic acid (C22:5n3)	0.19 ± 0.08 ^A^	0.15 ± 0.05 ^B^

Means with different superscripts within the same line represented differ significantly at *p* < 0.01 (A, B) or *p* < 0.05 (a, b) level.

**Table 5 animals-13-00965-t005:** The average of fatty acids and their groups according to SCC groups.

FA, g/100 g of Total FA	SCC Group (10 × 1000 cell/mL)		
	Low	Medium	High
	<750	750–1000	>1000
	N = 60	N = 30	N = 45
C4:0	1.17 ± 0.84 ^A^	1.72 ± 0.96 ^B^	0.88 ± 0.45 ^A^
C6:0	1.03 ± 0.18 ^a^	1.12 ± 0.17 ^a^	0.95 ± 0.19 ^b^
C8:0	1.96 ± 0.34 ^ab^	2.04 ± 0.52 ^a^	1.79 ± 0.45 ^b^
C10:0	7.93 ± 1.49	8.35 ± 2.19	7.52 ± 2.01
C11:0	0.22 ± 0.10	0.22 ± 0.07	0.20 ± 0.05
C12:0	3.90 ± 1.06	4.00 ± 0.97	3.53 ± 1.16
C13:0	0.15 ± 0.13 ^ab^	0.12 ± 0.10 ^a^	0.18 ± 0.09 ^b^
C14:0	9.51 ± 1.43	9.93 ± 1.60	9.18 ± 1.44
iC14:0	0.47 ± 0.22 ^A^	0.50 ± 0.19 ^AB^	0.58 ± 0.14 ^B^
C14:1n7	0.18 ± 0.12	0.20 ± 0.11	0.16 ± 0.22
C15:0	0.87 ± 0.16	0.86 ± 0.13	0.95 ± 0.40
iC15:0	0.08 ± 0.10 ^AB^	0.11 ± 0.07 ^A^	0.05 ± 0.08 ^B^
aC15:0	0.24 ± 0.13 ^A^	0.24 ± 0.02 ^A^	0.32 ± 0.15 ^B^
C16:0	26.26 ± 2.63	26.56 ± 3.11	27.62 ± 3.90
C16:1n9 t	0.59 ± 0.19 ^a^	0.49 ± 0.06 ^b^	0.59 ± 0.16 ^a^
C16:1n9	0.91 ± 0.13 ^A^	0.78 ± 0.13 ^B^	0.92 ± 0.18 ^A^
C16:1n7	0.67 ± 0.22 ^a^	0.60 ± 0.15 ^ab^	0.58 ± 0.10 ^b^
C17:0	0.90 ± 0.28 ^A^	0.67 ± 0.17 ^B^	1.02 ± 0.32 ^A^
iC17:0	0.11 ± 0.11 ^A^	0.20 ± 0.14 ^B^	0.12 ± 0.07 ^A^
C17:1n9	0.44 ± 0.22 ^a^	0.30 ± 0.07 ^b^	0.40 ± 0.13 ^a^
C18:0	10.60 ± 2.98	10.18 ± 1.51	10.76 ± 2.10
C18:1n9t	1.62 ± 0.76	1.74 ± 0.60	1.81 ± 0.69
C18:1n9	24.01 ± 4.14	23.22 ± 5.32	23.52 ± 4.29
C18:1n7	0.98 ± 0.26 ^a^	0.83 ± 0.27 ^b^	1.03 ± 0.19 ^a^
C18:2n6t	0.34 ± 0.16	0.40 ± 0.17	0.35 ± 0.14
C18:2n6c,t	0.22 ± 0.10	0.25 ± 0.10	0.21 ± 0.06
C18:2n6t,c	0.23 ± 0.13 ^AB^	0.19 ± 0.13 ^A^	0.28 ± 0.12 ^B^
C18:2n6	2.08 ± 0.57	1.95 ± 0.29	2.23 ± 0.58
C18:3n3	0.65 ± 0.30	0.64 ± 0.24	0.68 ± 0.23
C20:0	0.23 ± 0.09 ^A^	0.34 ± 0.17 ^B^	0.25 ± 0.05 ^A^
C20:1n9	0.08 ± 0.07 ^ab^	0.11 ± 0.06 ^a^	0.07 ± 0.04 ^b^
C20:4n6	0.16 ± 0.07 ^a^	0.14 ± 0.04 ^a^	0.19 ± 0.06 ^b^
C20:5n3	0.05 ± 0.06	0.02 ± 0.05	0.04 ± 0.05
C21:0	0.71 ± 0.78	0.67 ± 0.24	0.69 ± 0.31
C22:0	0.09 ± 0.12	0.10 ± 0.07	0.09 ± 0.05
C22:4n6	0.01 ± 0.06	0.00 ± 0.00	0.03 ± 0.08
C22:5n3	0.17 ± 0.08	0.16 ± 0.04	0.18 ± 0.08
C22:6n3	0.01 ± 0.02	0.00 ± 0.00	0.00 ± 0.00

Means with different superscripts within the same line represented differ significantly at *p* < 0.01 (A, B) or *p* < 0.05 (a, b) level.

## Data Availability

The data presented in this study are available within the article.

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
