# Peer review of "Associations between Somatic Cell Count and Milk Fatty Acid and Amino Acid Profile in Alpine and Saanen Goat Breeds"

_animals, 2023, doi:10.3390/ani13060965_

Round 1
Reviewer 1 Report
Rows 137 – 139: Not well understandable sentence.
Table 2: It would be useful to insert an extra line with the number of cows in the different groups
Row 160: all investigated goats did not have visual symptomps of mastitis à none of the investigated goats had visual symptomps of mastitis
Figure 1 consists of 2 figures. It would be better to be referred as figure 1 and figure 2. Then the subsequent figures should also be re-numbered.
Table 2: Notation of differences with letters is not well understandable this way. It is not written if numbers with the same or with different letters are significant. (e.g. SCC 1085 and 2201 is not different, as both marked with „a” ?)
Table 3: Again, it is a bit difficult to follow the differences and their levels.
Others:
- In the article, goats are referred as being different from cattle due to their apocrine milk secreting cells. However, cows also have apocrine milk secretion cells.
- Were the differences in milk composition of different SCC groups not investigated by breeds? As it can be seen, there was a significant difference between SCC of Alpine and Saanen breeds, and also, in protein content. Is it not possible that high SCC group had e.g. higher portein content because there were mostly goats of Alpine breed? To exclude this, at least the number of goats of the two breeds within each SCC group should be shown, I think.
It is not written if data of the different parameters showed a normal distribution or not.
Under the figures the editing of the text differs from the other parts.
Author Response
Thank You for the comments and Suggestions. We corrected manuscript according your comments and suggestions. Cover letter you will find in attached document.

Reviewer 2 Report
Dear authors,
The topic of the present paper is very important considering that description of goat milk characteristics is essential for improve the quality and has more diagnostic tools againts mastitis.
However, there are some comments and observations to attend. Please check the file

Author Response

(The authors gave the same response as above.)

Reviewer 3 Report
1- Authors must cite the protocol number of Animal Care and Use Committee;
2- In the Material and methods, the authors should specify the average weight of animals and day in milk in (days);
3- The dry matter intake of the goats is an important information and should not be omitted;
4- How often the animals were fed?
Discussion
The higher content of fat in Alpine may be related to lesser milk production in this breed this could be the main factor no specifically to SCC, please include this factor of fat variation in your discussion.
Author Response

(The authors gave the same response as above.)

Reviewer 4 Report
The subject of the study is very interesting and worth publishing. However, the manuscript has some flaws in the material and results description, thus a major revision is needed.
First of all organizing of the manuscript should be changed as follow:
Full title, Authors, Authors' affiliations including department and post/zip codes, Corresponding author, Abstract, Keywords, Implications, Introduction, Material and methods, Results, Discussion, Ethics approval, Data and model availability statement, Author ORCIDs, Author contributions, Declaration of interest, Acknowledgements, Financial support statement, References, Tables, List of figure captions. Some of them are lacking in the presented manuscript. Moreover, the style of references showing should be taken into account.
The substantive comments:
1. L60-61. Apocrine secretion of goat milk – milk fat is excreted in an apocrine way in all mammalian species. Regarding the way of protein secretion, Neveu et al. (2002) wrote: “We suggest that the apocrine pathway of secretion described in the goat could be the consequence of the dysfunction observed in the intracellular transport of caseins when alpha S1-casein is lacking.” Is alpha S1 casein lacking in Lithuanian goats?
Neveu, C., Riaublanc, A., Miranda, G., Chich, J. F., & Martin, P. (2002). Is the apocrine milk secretion process observed in the goat species rooted in the perturbation of the intracellular transport mechanism induced by defective alleles at the $\alpha_ {{\rm s} 1} $-Cn locus?. Reproduction Nutrition Development, 42(2), 163-172.
2. Material and methods – the animal material needs to be described in more detail. Namely, the frequency of goats in each group (breed, SCC, lactation) should be presented. Next, an explanation, of why the observation was conducted only in the mid of lactation is needed. The authors wrote that milk samples were collected from June 1, but then they informed that sampling was on the 20th day of each month. What was happening between June 1 and June 2
3. Why the microbiological analysis was not done?
4. Please, add information in the Introduction chapter about which amino acids are absent in goat milk
5. Statistical analysis – there were three effects that could influence the traits. Are all of them in the model? Were also interactions between them analyzed? The extracted results of the breed x SCC groups interaction would be more accurate for both effects.
6. Is the analysis of SCC as traits using the SCC group as the effect useful and informative?
7. All tables – the information that values (with units if the unit is the same for all traits) are presented in the tables should be written in the title. I recommend changing the way of significance marking (in the figures, too) – please, use capital and small letters (A, B, and a,b) to differentiate between p-values. Probably the differences between SCC groups in table 2 are marked incorrectly – I understand that there are no differences between all three groups.
8. L165-166 – To my knowledge, the cytoplasmic particles are not counted by commonly used devices, nowadays because the basing method is the staining of the whole cells (DNA staining). Is it not the truth?
9. Tables 3 and 4 – there is enough space to put the whole names of amino acids and fatty acids
10. Conclusion – in this Chapter only conclusions should be presented, thus, L307-319 should be moved to the end of the Discussion chapter as the summing up of the results. Please explain, what these biomarkers can be used for.
Author Response

(The authors gave the same response as above.)

Author Response

(The authors gave the same response as above.)

Round 2
Reviewer 4 Report
The authors corrected the manuscript according to almost all of my remarks. I have only a few minor remarks.
L99-102 and Table 2 - I recommend using "N" (capital letter) instead of "n" as usually, "n" is an attribute of a large population while here we have only a small sample from a population.
Please correct the order of figures 1 and 2. Figure 1 should be cited first in the text, so please place figure 2 as the first one and renumber both of them.
I can see that we did not understand each other about the signification marking. I recommended using only letters, without stars. As a rule, small letters a, b,c... mean the significance at p<0.05 while capital letters A, B... mean significance at p<0.01. I can see the authors want to mark also significance at p<0.001 while it is not necessary, in my opinion. Especially SCFA in figure 4 is very difficult to understand after correction. You can add one more column with the p-value.
Please see for example Zhang, S., Jiang, E., Wang, K., Zhang, Y., Yan, H., Qu, L., ... & Pan, C. (2019). Two insertion/deletion variants within SPAG17 gene are associated with goat body measurement traits. Animals, 9(6), 379.
Author Response

(The authors gave the same response as above.)

Reviewer 5 Report
Thank you, the reprimands have been taken into account and clarifications have been made.
However, the conclusions could be broader, providing an answer to the stated objective of the paper.
Although the duration of the study was 3 months, the authors performed an in-depth analysis of goat's milk, determining the amount of fatty acids and amino acids in the milk and, as far as possible, compared it with the results of other authors' research. I would not mind publishing the work.
Author Response
Thank You for the comments and Suggestions. Cover letter you will find in attached document.
